# Patient preferences when searching for clinical trials and adherence of study records to ClinicalTrials.gov guidance in key registry data fields

**Thomas M. Schindler** [1]*, **Frank Grieger**[2], **Anna Zak**[3], **Ramona Rorig**[4], **Kavya Chowdary Konka**[5], **Anna Ellsworth**[6], **Christopher Pfitzer** [7], **Keir Hodge**[8], **Christine Crandall** [9], on behalf of the TransCelerate Clinical Research Access & Information Exchange Initiative[¶]

1 Medical Writing Europe, Boehringer Ingelheim Pharma, Biberach, Germany, 2 Data Disclosure Governance & Standards, UCB BioSciences GmbH, Monheim, Germany, 3 Clinical Data Disclosure & Transparency, Merck Sharp & Dohme Corp., Rahway, New Jersey, United States of America, 4 Clinical Trial Disclosure, Medical and Development, Astellas Pharma, Northbrook, Illinois, United States of America, 5 Trial Disclosure, AbbVie Inc., North Chicago, Illinois, United States of America, 6 UCB Biosciences Inc, Raleigh, North Carolina, United States of America, 7 Transparency Operations, UCB BioSciences, Raleigh, North Carolina, United States of America, 8 PDG Neuroscience, F. Hoffmann-La Roche Ltd, Basel, Switzerland, 9 GSK, Clinical Project Management, Collegeville, Pennsylvania, United States of America

¶ A list of other current and previous members of the TransCelerate Clinical Research Access & Information Exchange Initiative can be found in the Acknowledgements.

* thomas.schindler@boehringer-ingelheim.com

**Data Availability Statement:** All relevant data are within the manuscript and its Supporting Information files.

## Abstract

ClinicalTrials.gov was started with the intention to create a consumer-friendly database for patients and others in search of information on clinical trials. However, there is no research on whether the content of ClinicalTrials.gov aligns with patient preferences. The TransCelerate Clinical Research Access & Information Exchange Initiative convened patient advisory boards and conducted a global online survey (N = 1070) to determine patient preferences when searching for clinical trials for participation. Patient feedback and ClinicalTrials.gov guidance documents were used to construct instruments to assess patient focus and guidance adherence of the Brief Title (a short lay title of the clinical trial) and Brief Summary (a high-level summary of study features) data fields in a representative sample (N = 346) of ClinicalTrials.gov records of interventional trials. When searching for clinical trials, survey participants rated condition (66.4%), trial location (57.0%), trial dates (52.9%), age and gender (48.6%), and health measurements (i.e., what the study measures) (45.5%) as the most important items. When presented with a list of trials from an initial search, participants saw condition, brief summary, study drug name, and brief title as the most helpful items. In a Brief Title, they wanted condition, health measurements, participant age, and study drug name. For Brief Summaries, participants preferred additional information on treatment duration, condition, study goal, health measurements, and frequency of visits. The assessment of patient focus in a representative sample of current ClinicalTrials.gov records showed that patient focus was underdeveloped as study records achieved only 52% (brief titles) and 50% (brief summaries) of the best possible score. The analysis of adherence to

**Funding:** TransCelerate BioPharma Inc. is a non-profit organisation with a mission to collaborate across the biopharmaceutical research and development community. The work within TransCelerate is performed by a number of initiatives. These initiatives develop practical solutions to overcome common challenges in clinical development and are drawn from the combined experiences of the member companies and various industry organizations. All authors and contributors are employees of individual companies from which they received remuneration. The authors and contributors were delegated by their employers to be members of the TransCelerate Clinical Research Access and Information Exchange Initiative. The funders provided support in the form of salaries for authors [TMS - Boehringer-Ingelheim Pharma, FG - UCB BioSciences, AZ - Merck Sharp & Dohme Corp., RR - Astellas Pharma, KCK - AbbVie Inc., AE - UCB BioScience, CP - UCB BioScience, KH - F. Hoffmann-La Roche Ltd, CC - GSK], but did not have any additional role in the study design, data collection and analysis, decision to publish, or preparation of the manuscript. The specific roles of these authors are articulated in the 'author contributions' section. TransCelerate served as a platform and provided project management and administrative support but had no role in study design, data collection and analysis, decision to publish, or preparation of the manuscript.

**Competing interests:** All authors are employed by a pharmaceutical company as indicated in by their affiliation [TMS -Boehringer-Ingelheim Pharma, FG - UCB BioSciences, AZ - Merck Sharp & Dohme Corp., RR - Astellas Pharma, KCK - AbbVie Inc., AE - UCB BioScience, CP - UCB BioScience, KH - F. Hoffmann-La Roche Ltd, CC - GSK]. In addition, all authors were members of the TransCelerate Clinical Research Access and Information Exchange Initiative. None of the authors have any additional role or function in TransCelerate outside of the Clinical Research Access and Information Exchange Initiative. No author received any additional remuneration for participating in this research. None of the pharmaceutical companies nor TransCelerate had any role in the study design; collection, analysis, and interpretation of data; writing of the paper; and/or decision to submit for publication. This does not alter our adherence to PLOS ONE policies on sharing data and materials. None of authors has any other competing interest. The article, its figures, and Supporting Information files are licensed under a Creative Commons Attribution (CC BY 4.0) license and in care of TransCelerate.

ClinicalTrials.gov guidance showed better scores (brief titles 69%, brief summaries 66%). We identified key information elements for registry users when evaluating clinical trials for participation. We found that aspects of patient focus are not common in current Clinical-Trials.gov entries. To support more user-friendly study records, we developed a tool to assess the quality of the plain language fields in study records prior to submission.

## Introduction

From the outset, the key purpose of ClinicalTrials.gov was to make information on clinical trials available to the public, to "individuals with serious or life-threatening diseases or conditions, to other members of the public, to health care providers, and to researchers" [1]. Hosted by the National Library of Medicine, ClinicalTrials.gov was to be a "consumer-friendly" database that provides easy access to information about clinical trials for patients and families and members of the public [2]. In 2007, the FDAAA widened the scope of ClinicalTrials.gov but retained the original purpose. The mandatory registration of clinical trials was complemented with the requirement to make trial results available [3, 4]. Currently (August 2019), Clinical-Trials.gov contains some 314,000 trials in 209 countries and has some 145,000 unique visitors daily (ClinicalTrials.gov website accessed 14 August 2019). As the most comprehensive clinical trial database, ClinicalTrials.gov is the key source for many other registries around the globe.

ClinicalTrials.gov presents the information on clinical trials in a structured way using different tabs and tabular data displays. Sponsors enter information either in free-text fields or choose from pre-defined drop-down menus. There is extensive guidance from ClinicalTrials. gov on the way sponsors should populate the different database fields.

While timeliness and comprehensiveness of clinical trial registration and results posting by sponsors have received a lot of attention [5–10], there is no research on the adequacy of the information provided in ClinicalTrials.gov from a patient and registry user point of view. We have been unable to identify any research publication that examines whether the information provided by ClinicalTrials.gov coincides with the preferences of patients and other members of the public when searching for information about clinical trials.

The mission of the TransCelerate Clinical Research Access & Information Exchange Initiative is to improve access to information about clinical research and clinical trial options. In 2018, TransCelerate formed a working group within the Initiative to investigate the barriers that may prevent patients and others from finding available clinical trials. The availability of pertinent information in ClinicalTrials.gov and other public registries is critical for people considering clinical trial participation.

Our research focused on two data fields: The Brief Title (a short title describing the trial) and the Brief Summary (a summary that provides a general, high-level overview of the trial); both are very prominent elements of each study record. They are shown on the first (Brief Title) and second (Brief Summary) result pages that are returned by ClinicalTrials.gov subsequent to a search request. As per ClinicalTrials.gov guidance [11, 12], the text that sponsors enter in these data fields needs to be in lay language, i.e., it needs to be understandable for the public. Since both data fields are free-text fields they offer the opportunity to sponsors to provide the information in a patient-focused way.

The overall objective of this research was to obtain insight into the information preferences of people using a clinical trial registry in search for a clinical trial. In addition, we wanted to

determine to what extent the resulting preferences are seen in a representative sample of current ClinicalTrials.gov entries. In detail we aimed at:

- Obtaining insight on the patients' preferences when evaluating study records for participation during the initial search and during refined evaluation.

- Determining which information items patients find helpful in brief titles and in brief summaries.

- Determining the extent to which the current entries in ClinicalTrials.gov reflect the patients' information preferences.

- Determining the extent to which current ClinicalTrials.gov records adhere to available NIH data submission guidance.

The insights gained were used in the development of a tool that helps sponsors to assess the quality of their clinical trial submissions to ClinicalTrials.gov. The tool is available at the TransCelerate homepage (https://transceleratebiopharmainc.com/clinical-research-access-information-exchange-assets/).

## Methods

We conducted Patient Advisory Board (PAB) Meetings and a global online survey to obtain insight into the information preferences of people using a clinical trial registry in search for a clinical trial. Based on the patients' feedback and on the survey results, we drew a representative sample of current ClinicalTrial.gov entries and analyzed the extent to which they meet information preferences of people searching for a clinical trial for participation. In addition, we investigated the adherence of current records for brief titles and brief summaries to the guidance [11, 12] provided by ClinicalTrials.gov.

### Patient advisory boards (PABs)

To develop an initial insight into patient preferences when using trial registries for identifying suitable trials, we commissioned the Center for Information and Study on Clinical Research Participation (CISCRP) to convene and organize a PAB in June (in-person in EU) and November (in-person in US) of 2018. Fifteen patients and patient advocates, (8 from EU and 7 from US) participated. The patients had all either participated in a clinical trial, had a chronic condition or were patient advocates. All were required to be fluent speakers of English. PAB members were compensated for their participation (travel costs, time).

### Global online survey

Based on the feedback from the PABs, a 20-question survey was developed. Survey participants were asked to rank or select various categories of information in regard to their helpfulness when searching for and then evaluating clinical trials in a registry for participation. The survey was conducted online using SurveyGizmo (https://www.surveygizmo.com/) for 5 weeks in February to March 2019 and was supported by CISCRP outreach efforts across their patient and caregiver communities. The survey was available in English, Japanese, German, and Spanish. Survey takers were compensated for their time by a mini-incentive (mail order voucher of $10). A copy of the survey questions and the full dataset is available in S1 Appendix.

## Assessment of current ClinicalTrials.gov records

Based on feedback from PABs and the online survey, we investigated the status of two data fields of current ClinicalTrials.gov records for patient focus and adherence to guidelines. We focused on brief title and brief summary because of their prominence in the ClinicalTrials.gov search result display, their importance for patients and members of the public, and the fact that these data fields are amenable to improvement because they permit text entries by the sponsor.

**Sample selection and data extraction.**   We drew a representative sample of ClinicalTrials.gov study records by downloading all records of all Phase II, III, or II/III clinical trials registered between 18 April 2017 and 17 April 2018. These dates were chosen to provide a sample that reflects current sponsor behavior post implementation of the Final Rule (FDAAA 801) [3, 4]. We limited the search to interventional studies (excluding devices), irrespective of sponsor type and recruitment status. Data were downloaded from ClinicalTrials.gov on 23 January 2019 and converted into an MS Excel file. From the overall sample (N = 3986), we randomly (MS Excel Rand function) selected 346 studies for detailed analysis. Our subset of 346 studies was fully representative of the ClinicalTrials.gov entries and comprised 8.7% of all entries at the time. The maximum difference between the two sets for sponsor type, clinical phase, initiation date, and drug vs other factors was 2.3%. Sample size calculation was done using nQuery Advisor Version 7.0.

The subset of study records (N = 346) was analyzed manually according to a set of criteria. Presence or absence of elements–including both content and format items—were coded to a numeric value that was then entered into an MS Excel file. All items were weighed equally. To be able to compare the assessments (S2 Appendix), we rescaled them to the same scale. All analyses are considered exploratory; p-values are for illustration only. There was no consideration of multiple testing.

**Assessment of patient focus of brief titles and brief summaries.**   We constructed two assessment instruments to assess patient focus, one for the analysis of brief titles and one for the analysis of brief summaries. The criteria used to assess patient focus were derived from the feedback in the PABs and the results of the global online survey. The assessment included the items that were selected by over 30% of the survey respondents when asked which information they considered helpful in brief titles and brief summaries when searching for clinical trials.

The instrument for brief titles considered the presence (or absence) of condition, health measurement, participant age, study drug name, and form of drug (pill or injection) based on their ranking in the online survey. The criteria "absence of technical terms" and "absence of unexplained abbreviations" were added based on feedback from the PABs and to reflect the fact that brief titles should be in lay language according to ClinicalTrials.gov requirements. All criteria were equally weighted; thus, the maximum (best possible) score was 7 for a brief title fulfilling all criteria.

The instrument to assess brief summaries included the information items considered important by more than 30% of survey participants. These were treatment duration, condition, study goal, health measurement, and frequency of visits. In addition, we included criteria provided as feedback from the PABs, namely that the information should be provided in full sentences, abbreviations should be defined, and the text be structured by short paragraphs or bullet points. All criteria had an equal weight and the maximum (best possible) score was 8.

**Assessment of adherence to ClinicalTrials.gov guidance of brief titles and brief summaries.**   The criteria used to assess adherence to the ClinicalTrials.gov guidance were derived from the most current versions of the Protocol Registration Data Element Definitions [11] and the ClinicalTrials.gov Protocol Registration and Document Upload Quality Control Review

Criteria [12]. The guidance comprises both items related to content and related to format. Both data fields need to be in lay language and certain elements need to be mentioned. For brief titles: participants, condition, intervention (study drug name) and for brief summaries: full sentences, study hypothesis/purpose, no references, statement on availability of expanded access.

For the analysis of the brief title data field, the requirement to mention participants was operationalized as presence of any humanizing term (namely presence of at least one of the following terms: patient, subject, participant, or volunteer) or mentioning participant gender or participant age. The requirement to be in lay language was operationalized such that brief titles and brief summaries should not contain technical study design terms or unexplained abbreviations. Further details of the evaluation rules are provided in S2 Appendix.

Hence, the instrument used to assess adherence of brief titles included 7 items: condition, study drug name, absence of technical study design terms, absence of unexplained abbreviations, absence of a period, presence of any humanizing term (participant, patient, subject, volunteer) or participant age or participant gender. In addition, the presence of the term "participant" was counted separately because the use of this term is emphasized by the guidance. The maximum (best possible) score was 7.

ClinicalTrials.gov provides little guidance for the brief summary field although it permits considerable length (5000 characters). Therefore, the instrument for assessment of adherence of brief summaries comprised only four items: absence of bibliographic references, complete sentences, study goal (hypothesis/purpose), and statement about the availability of expanded access programs. To compare the adherence assessment with the patient focus assessment each item was multiplied by 2 resulting in a maximum value of 8.

**Validation of assessments.**   The patient focus assessment criteria were qualitatively validated with the PAB members. Examples of brief titles and brief summaries with the highest and lowest values were presented for rating and assessment and discussion. PAB members generally rated high-performing titles and summaries as being more understandable and more helpful than low-performing ones.

To further validate the assessments of brief titles and brief summaries, a random 10% sample of the subset was independently re-assessed by two team members and their results were compared to the initial assessments. The concordance between the initial and reassessment was >90% for all the elements for the brief title. For brief summary, the concordance between initial and reassessment was >85% for all items.

## Statistical analysis

Statistical analyses were performed using SAS software (version 9.4; SAS Institute, Cary, NC, USA).

## Results

### Patient advisory board (PAB) meetings

Participants articulated awareness of clinical trials as options and felt registries were important for searching and identifying clinical trials for which they may be eligible. Most participants were familiar with and had used registries to identify clinical trials. Those who had searched for clinical trials did so using general internet searches and searches in registries but felt these sources were difficult to navigate and fell short of expectations with regard to usefulness, personalization, and clarity. Participants identified condition (i.e., disease being studied) as an important item when searching for a clinical trial, primarily because this determines whether the trial is relevant to them. Location of the investigational site and information in a brief title

and a brief summary were also important for participants. In addition, participants expressed appreciation of trial descriptions that explained the level of their commitment when participating, e.g., the number of visits, the planned treatment duration for individual patients, and the health measurements (i.e., what the study measures). Participants emphasized that the information should be written in language that is easy to understand and the text should be easy to read, i.e., it should not be presented as one block but should be structured by bullets or split into several short paragraphs.

## Global online survey

**Demographics.** In total, 1070 people from 28 countries responded to the online survey. About a quarter of respondents resided in the USA (26%), 21% lived in Japan, 13% in Australia, 11% in the United Kingdom, 10% in Germany, 8% in Spain, and 7% in Mexico. Only a minority (11%) had participated in a clinical trial but 22% had visited a registry. ClinicalTrials.gov was the most visited registry (51%) followed by pharmaceutical company registries (39%). Demographic characteristics of the survey participants are provided in Table 1.

**Patient preferences.** Survey takers were asked to imagine visiting a clinical trial registry website with the intention to search for clinical trials in which to participate. They were presented with a list of 12 items and asked to select the 5 most helpful items in a study record (Table 2).

**Table 1. Demographics of 1070 survey respondents.**

| Characteristic | Percent of Subjects[a] |
|---|---|
| **Gender** | |
| Female | 52% |
| Male | 48% |
| Other, including "prefer not to answer" | 0.4% |
| **Mean age (SD), years** | 45.3 (14.2) |
| **Education** | |
| High school/secondary education (grades 9–12, no degree) | 9% |
| High school/secondary education graduate (or equivalent) | 16% |
| Technical or trade school training | 13% |
| Some college/higher education (1–4 years, no degree) | 12% |
| Associate degree (incl. occupational or academic degrees) | 7% |
| Bachelor's degree | 23% |
| Master's degree | 11% |
| Professional school degree or doctorate | 8% |
| **Employment** | |
| Full-time/self employed | 50% |
| Part-time | 13% |
| Retired | 13% |
| Homemaker/Stay at Home | 11% |
| Unemployed/looking for work | 8% |
| Student | 3% |
| Other | 3% |

[a] Percentages are rounded to the nearest whole number. The only item not provided in terms of percent of subjects is mean (standard deviation) age which is provided in years.

**Table 2. Most helpful items in a study record based on being identified among the top 5 items by at least 30% of survey respondents.**

| Short term | Description | Frequency[a] |
|---|---|---|
| Condition | Condition or disease being studied in the clinical trial, for example diabetes or heart disease | 66.4% |
| Location | The name, address, and contact information for each place where the clinical trial visits would take place | 57.0% |
| Dates | The dates the clinical trial will start and end | 52.9% |
| Age and gender | Age and gender of participants | 48.6% |
| Health measurements | Health measurements or observations examined to determine the effect from the clinical drug, for example the reduction of cholesterol, "what the study measures" | 45.5% |
| Study drug | Name of clinical trial drug being studied | 42.5% |
| Study type | Study type, e.g., interventional, observational clinical study, or expanded access[b] | 38.4% |
| Enrolment status | Whether the clinical trial is actively enrolling | 38.2% |
| Clinical phase | Phase of the clinical trial, e.g., Phase I, II, III[b] | 32.4% |

[a] Percent of respondents who considered the item to be among the top 5 most helpful items.

[b] Detailed descriptions were provided as hover-overs.

Survey participants were then to imagine that the registry had returned a list of trials in response to their initial search. They were provided with a list of nine items representing free-text fields in ClinicalTrials.gov. Their task was to rank these items from most to least helpful (Table 3).

Condition, brief summary, study drug, and brief title were the most important items when evaluating a list of trials for potential participation. This result was in line with the feedback from the PABs.

**Table 3. Information items ranked most (#1) to least helpful (#9) by survey respondents for selecting clinical trials from an initial list of trials.**

| Overall Rank | Short term | Description |
|---|---|---|
| 1 | Condition | Condition or disease being studied in the clinical trial, for example diabetes or heart disease |
| 2 | Brief Summary | A brief summary that provides a general overview of the clinical trial, written in plain language i.e. Language that can be easily understood and does not contain medical jargon |
| 3 | Study drug | Information about the drug that is being studied |
| 4 | Brief Title | A short title describing the clinical trial, written in plain language i.e. language that can be easily understood and does not contain medical jargon. |
| 5 | Eligibility Criteria | Criteria needed for participation, for example inclusion / exclusion criteria i.e. characteristics that possible study volunteers must have if they are to be included in the study. |
| 6 | Health measurements | Health measurements or observations examined to determine the effect from the clinical drug, for example the reduction of cholesterol, "what the study measures" |
| 7 | Location | The name, address, and contact information for each place where the clinical trial visits would take place |
| 8 | Placebo control | Chance of receiving the clinical trial drug or placebo |
| 9 | Sponsor information | The name, email address, and / or phone number of the individual, company, or research organization responsible for paying for the clinical trial |

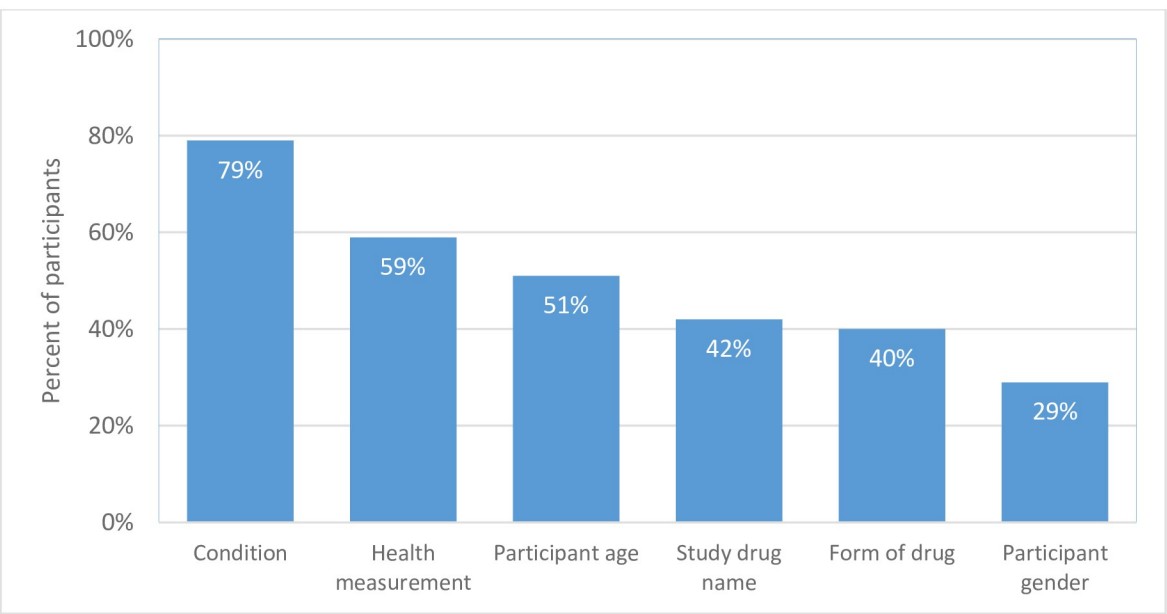

**Fig 1. Preferences for information items in brief titles.** Bars denote the percentage of participants who ranked the item as one of the three most helpful (percentage rounded to the nearest whole number). For definitions, refer to Table 2; participant age (e.g., children or over 60 years of age); form of drug (e.g., pill or liquid).

Two questions obtained further insight into the information that survey participants want to see in brief title and brief summaries. For brief titles, survey participants were asked to select the 3 most helpful pieces of information out of 6 items. Condition, health measurement, participant age, drug name, and drug form (pill or injection) were chosen by at least 30% of participants and were thus identified as the most important information in the brief title data field (Fig 1).

For the brief summary, survey participants were asked to select pieces of information that they would find helpful when considering participation in a trial in addition to those provided in the brief title. The duration of treatment, detailed condition description, goal of the trial, health measurements, and frequency of visits were the most frequently selected items (Fig 2). Interestingly, survey participants expressed little interest in whether they could be receiving placebo (treatment allocation) and the total number of participants (Fig 2).

### Patient focus assessment of brief title and brief summary in current ClinicalTrials.gov records

**Patient focus assessment of brief titles.** Almost all records contained a term describing the condition being studied and the name of the drug that was evaluated. The majority of brief titles was free of technical study design terms and did not contain unexplained abbreviations. Overall, 44.8% of brief titles contained the condition, the study drug name, and information on the participants. In our representative sample of 346 trials, the mean value for patient focus was 3.66 (SD 0.85) representing 52% of the maximum value (7). Hence, there is substantial room for improvement in patient focus for brief titles for study records (Table 4).

**Patient focus assessment of brief summaries.** Overall, only 13.9% of brief summaries included all of the top three information items identified by the survey (treatment duration, condition, goal of trial). However, condition and goal of the trial were included in most brief summaries (70.5%). Treatment duration, which was ranked 1st in the survey, was included in

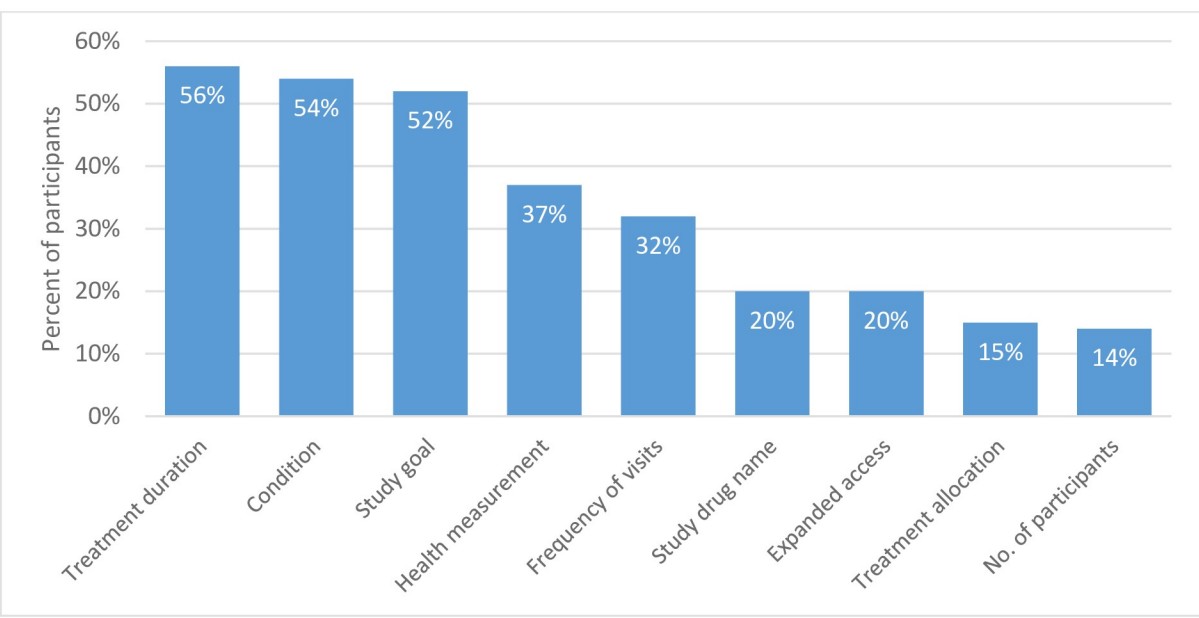

**Fig 2. Preferences for information items in brief summaries.** Bars denote the percentage of participants who ranked the item as helpful (percentage rounded to the nearest whole number). For other definitions, see Table 2; treatment duration is how long the study drug will be given (e.g., one month or one year); expanded access is a statement on the availability of expanded access; treatment allocation is a description of the number of participant groups that receive study drug or placebo.

only 19.1% of brief summaries signifying a disconnect with preferences expressed by patients and survey respondents. Also, format features such as paragraphs and bullets that would help the general public reading brief summaries were only infrequently used (Table 5). Based on our representative sample of 346 studies, the mean value for patient focus for brief summaries was 4.03 (SD 1.29) representing 50.4% of the maximum (8).

## ClinicalTrials.gov guidance adherence assessment of brief title and brief summary

As a second aspect of quality of records in ClinicalTrials.gov, we investigated the adherence to ClinicalTrials.gov guidance for brief titles and brief summaries. Almost all brief titles in our sample did comply with the guidance not to include a period at the end and a great majority

**Table 4. Assessment of patient focus in brief titles: Presence of specified elements in a representative sample of 346 current ClinicalTrials.gov study records.**

| Criterion | Criterion Met | |
|---|---|---|
| | Number (%) of Records | 95% CI |
| Condition | 329 (95.1) | (92.2, 97.1) |
| Study drug name | 301 (87.0) | (83.0, 90.4) |
| Absence of technical study design terms | 278 (80.3) | (75.8, 84.4) |
| Absence of unexplained abbreviations | 260 (75.1) | (70.2, 79.6) |
| Participant age | 39 (11.3) | (8.1, 15.1) |
| Health measurements (what the study measures) | 35 (10.1) | (7.1, 13.8) |
| Study drug form (pill or liquid) | 23 (6.6) | (4.3, 9.8) |

**Table 5. Assessment of patient focus in brief summaries: Presence of specified elements in a representative sample of 346 current ClinicalTrials.gov study records.**

| Criterion | Criterion Met | |
|---|---|---|
| | **Number (%) of Records** | **95% CI** |
| Condition | 321 (92.8) | (89.5, 95.3) |
| Complete sentences | 316 (91.3) | (87.9, 94.1) |
| Study goal | 263 (76.0) | (71.2, 80.4) |
| Absence of unexplained abbreviations | 234 (67.6) | (62.4, 72.5) |
| Health measurements | 96 (27.7) | (23.1, 32.8) |
| Paragraph format used | 74 (21.4) | (17.2, 26.1) |
| Treatment duration | 66 (19.1) | (15.1, 23.6) |
| Bullet points | 20 (5.8) | (3.6, 8.8) |
| Frequency of visits | 15 (4.3) | (2.4, 7.0) |

included the condition and the study drug name. However, despite the requirement to be understandable for the lay public, a significant proportion of brief titles contained technical study design terms (19.7%) or abbreviations that were not explained (24.9%) (Table 6).

The mean adherence assessment score was 4.86 (SD 0.94), representing 69.4% of the maximum of 7. Only 10 of the records (2.9%) met all criteria included in the adherence assessment instrument.

**Guidance adherence assessment of brief summaries (Table 7).** Most records did not include references and were written in complete sentences, the study goal was provided in ¾ of all records. No study in our representative sample mentioned whether expanded access was available to study participants. Overall, 69.9% fulfilled three criteria (no references, complete sentences, study goal). The mean value for guidance adherence of brief summaries was 5.3 (SD 1.13) representing 66.5% of the maximum (best possible) score.

In summary, the analyses of adherence to guidance for current brief title and brief summary records shows that on average sponsors adhere only to a limited extent to available ClinicalTrials.gov guidance.

**Table 6. Assessment of adherence to ClinicalTrials.gov guidance in brief titles: Presence of specified elements in a representative sample of 346 current clinicaltrials.gov study records.**

| Criterion | Criterion Met | |
|---|---|---|
| | **Number (%) of Records** | **95% CI** |
| Absence of period at end of title | 334 (96.5) | (94.0, 98.2) |
| Condition | 329 (95.1) | (92.2, 97.1) |
| Study drug name | 301 (87.0) | (83.0, 90.4) |
| Absence of technical study design terms | 278 (80.3) | (75.8, 84.4) |
| Absence of unexplained abbreviations | 260 (75.1) | (70.2, 79.6) |
| Term 'Patient' [a] | 104 (30.1) | (25.3, 35.2) |
| Participant age [a] | 39 (11.3) | (8.1, 15.1) |
| Term 'Subject' [a] | 26 (7.5) | (5.0, 10.8) |
| Term 'Participant' [a] | 13 (3.8) | (2.0, 6.3) |
| Participant gender [a] | 8 (2.3) | (1.0, 4.5) |
| Term 'Volunteer' [a] | 1 (0.3) | (0.0, 1.6) |

[a] For the calculation of the overall value, all of these items were combined in one category; "participant" was additionally counted as individual item.

**Table 7. Assessment of adherence to clinicaltrials.gov guidance in brief summaries: Presence of specified elements in a representative sample of 346 current ClinicalTrials.gov study records.**

| Criterion | Criterion Met | |
|---|---|---|
| | Number (%) of Records | 95% CI |
| Absence of bibliographic references | 340 (98.3) | (96.3, 99.4) |
| Complete sentences | 316 (91.3) | (87.9, 94.1) |
| Study goal /purpose | 263 (76.0) | (71.2, 80.4) |
| Availability of expanded access[a] | 0 (0.0) | (0.0, 1.1) |

[a]None of the brief summaries in our sample mentioned the availability of expanded access although the guidance indicated that this was required. In a post-analysis conversation, NIH pointed out that they had mistakenly included the request for information on expanded access into their guidance documents for all trials.

**Comparing patient focus and adherence assessments in current ClinicalTrials.gov study records.** For both data fields, the average values for patient focus were comparatively low and in a similar range (52% and 50% of the maximum) while the average values for adherence were higher (66% to 69% of maximum). The mean differences between mean patient focus and mean adherence values were comparable for both data fields. This indicates that current study records are more successful at fulfilling ClinicalTrials.gov requirements than at providing information needed by potential trial participants (Table 8).

We checked whether the assessments were influenced by the type of sponsor (pharma industry vs. academia), the phase of the study (II vs. III), the study initiation date (2017 vs. 2018), and study type (drug vs. non-drug, e.g. surgery or nutritional supplement). We found that on average industry funded trials were likely to perform slightly higher on the brief title patient focus assessment (industry 3.74, non-industry 3.56, mean difference 0.182, $p = 0.0477$) and the brief title adherence assessment (industry 5.10, non-industry 4.59, mean difference 0.51, $p < 0.0001$). For brief summary, none of the factors had any influence on the assessment.

In addition, we checked whether the assessment was influenced by the weighting of the individual items (equal weight of all elements). However, results did not change qualitatively when the weighting was done according to the percentage of survey respondents ranking the items as important (0–25% = 0.5; >25% - 50% = 1.0; >50% - 75% = 1.5; >75% = 2.0).

**Table 8. Comparison of patient focus and guidance adherence in a representative sample of 346 current ClinicalTrials.gov study records.**

| Information Item Assessment | Score | | | |
|---|---|---|---|---|
| | Mean (SD) | 95% CI | Percent of Maximum Score | Observed Range |
| **Brief title** | | | | |
| Patient focus (0–7) | 3.66 (0.85) | (3.57, 3.75) | 52% | 1 to 6 |
| Guidance adherence (0–7) | 4.86 (0.94) | (4.76, 4.96) | 69% | 2 to 7 |
| Mean difference[a] | 1.21 (0.73) | (1.13, 1.28) | - - | -1 to 3 |
| **Brief summary** | | | | |
| Patient focus (0–8) | 4.03 (1.29) | (3.90, 4.17) | 50% | 1 to 8 |
| Guidance adherence (0–8)[b] | 5.31 (1.13) | (5.19, 5.43) | 66% | 0 to 6 |
| Mean difference[a] | 1.28 (1.16) | (1.16, 1.40) | - - | -3 to 4 |

[a] Guidance adherence score minus patient focus score.

[b] Assessment tool had 4 items; thus, each item was worth 2 points instead of 1 to make the possible range of scores 0 to 8 (same as for the patient focus assessment tool).

## Discussion

ClinicalTrials.gov had been inaugurated to make it easier for patients, health care providers, and the public to find clinical trials with investigational drugs [1]. Its objective was to provide information to an audience that is unfamiliar with the clinical research methodology and associated terminology. This objective was maintained through all updates of ClinicalTrials.gov.

While there is a growing body of research on timeliness of registration, on posting of results, and on other quality aspects [5–10, 13, 14], there are no investigations on whether current ClinicalTrials.gov entries support the initial objective of this database.

We used Patient Advisory Boards (PAB) and a global online survey to determine the information preferences of patients and other registry users when searching for clinical trials for participation.

PAB participants identified brief title and brief summary as important when searching for clinical trials. For brief summaries, they emphasized the importance that information is provided in full grammatical sentences in a language that is easy to understand. In addition, the information should be provided in a structured way, i.e., that the text is broken down into short paragraphs or structured by bullets. The global survey identified condition, location of study sites, trial start and end dates, age and gender of trial participants, the health measurements (what the study measures), and the name of the study drug as the most important information items when searching for a clinical study for participation. This is in line with the results of a study on public attitudes towards clinical research that also determined health measurements, location of study sites, length of participation as among the most important factors for decisions on trial participation [13].

We focused on brief title and brief summary because of their prominence in the ClinicalTrials.gov search result display, their importance for patients and members of the public, and the fact that these data fields are amenable to improvement by the sponsor.

Based on the survey feedback, a brief title should mention the condition, health measurement, participant age, study drug name, and study drug form. To be most helpful, a brief summary should provide additional information on the treatment duration, condition, study goal, health measurements, and frequency of visits. For brief title, condition was the most important information item (79%) for the identification of a suitable study. However, due to the requirement to be in plain language and due to space limitations, the condition can only be expressed in broader terms in brief titles. For example, a brief title of a study in patients with stage IIIB/IV non-small lung cancer may only mention the term "lung cancer" in the brief title and provide the more detailed description of the condition in the brief summary.

For brief summaries, treatment duration (56%), condition (54%), and study goal (52%) were identified as the most helpful items. We hypothesize that "condition" was also ranked highly in brief summaries because survey participants had been asked to select those items that provide information in addition to what is presented in the brief title.

Our analysis of 'patient focus' and "guidance adherence" in a representative sample of ClinicalTrial.gov entries found that for both the brief titles and brief summaries, the mean value for patient focus was substantially lower (52% and 50% of the maximum [best possible] value) than the mean value for adherence to guidance (69% and 66% of the maximum value). This result is unlikely to be an artefact as the sample size in our analysis was similar to that used in other research investigating other aspects of trial registration [5] and our sample represented a similar proportion of ClinicalTrials.gov entries as used in research on other quality aspects of ClinicalTrials.gov records (our sample: 8.7% vs. sample in reference [5]: 5%).

Our findings indicate that the current state of registrations in ClinicalTrials.gov leaves much room for better alignment with patients' and registry users' preferences. Sponsors

should take the initiative to improve the quality of brief titles and brief summaries in their clinical trial registrations. We found that sponsors' study entries had better adherence to Clinical-Trials.gov guidance, albeit with room for improvement, than user friendliness. Overall, only about 70% of brief summaries had the three required components (no bibliographic references, complete sentences, and providing the study goal).

While adherence to ClinicalTrials.gov guidance is not optimal, the bigger issue is the widespread absence of patient focus in most brief titles and brief summaries. In our analysis, no brief title and only two brief summaries (0.6%) achieved the maximum assessment for patient focus. A reason for the apparent absence of patient focus in brief summaries could be that sponsors consider trial registration in ClinicalTrials.gov solely as a legal requirement and not as an opportunity to engage with patients and raise interest for their trials. The higher scores for adherence support this view.

Improvements of patient focus for brief titles and brief summaries could be achieved by either making the information items we have identified mandatory or by including them into the guidance documents for sponsors. However, the improvement of guidance needs to be complemented by making sponsors realize that ClinicalTrials.gov entries have a potential to much more adequately inform patients and the general public about clinical trials. More patient focused brief titles and brief summaries will allow for more informed decision making on clinical trial participation. The Clinical Research Access & Information Exchange Initiative has created a Clinical Trial Registration Tool that will help sponsors assess their data quality so they may produce more patient-focused, compliant clinical trial submissions. It is available on the TransCelerate website ((https://transceleratebiopharmainc.com/clinical-research-access-information-exchange-assets/)).

Our study has provided a basis for better understanding the patient preferences for entries in study records. Improving the patient focus of the brief titles and the brief summaries will help to improve the usefulness of Clinicaltrials.gov entries and thus fulfill its original intention.

## Limitations

The global web-based survey participants represent a convenience sample facilitated by CISCRP patient and caregiver outreach network and as such, findings may not be representative of the entire population. However, our survey appears to be the largest on this topic to date. The survey participants were provided with a range of possible answers for all questions. Although the options had been included based on feedback and discussion of the PABs, other items could have been important to patients that were not offered as potential answers. Despite the fact that the criteria selected for the patient focus instrument were based on feedback from the PABs and the results of the survey, the use of a cut-off (30%) for the criteria selected may fail the information needs of minorities.

## Conclusion

We have identified the brief title and brief summary together with information on the condition studied and study drug name as key elements for patients and other registry users when evaluating trials for participation. The results of a global patient survey allowed us to develop assessment instruments for patient focus and adherence to ClinicalTrials.gov guidance for brief title and brief summary. We found that study records had higher values for adherence to ClinicalTrials.gov guidance than for patient focus. Including the information items preferred by patients and registry users into guidance documents could support the original intention of ClinicalTrials.gov. This would benefit both sponsors (potential for more rapid recruitment) and the public (more accessible information on clinical trials). Sponsors need to take action

for making the evaluation of clinical trials easier for patients and other registry users and should more fully adhere to ClinicalTrials.gov guidance.

## Supporting information

**S1 Appendix. Global online survey questions and response summary.**
(PDF)

**S2 Appendix. Assessments of brief titles and brief summaries.**
(PDF)

## Acknowledgments

The authors gratefully acknowledge the support of TransCelerate BioPharma Inc, a nonprofit organization dedicated to improving the health of people around the world by accelerating and simplifying the research and development (R&D) of innovative new therapies, and the following members of the TransCelerate Clinical Research Access & Information Exchange Initiative: Munther Baara, Pfizer Inc.; Kimberly Doggett, UCB; Mohammed ElGuerche, Novartis; Isabelle Gautherot, UCB; Cathleen Jewell, AbbVie Inc.; Ranjita Mishra, Sanofi; Manoj Prabhu, Amgen; Christa Polidori, BMS; T.J. Sharpe, Patient Advisor; Lynes Torres, PA consulting; and Frances Pu, Renaissance Writing Services, for medical writing support. CISCRP (Center for Information & Study on Clinical Research Participation) is acknowledged for expert support in conducting the patient advisory boards and the online survey.

## Author Contributions

**Conceptualization:** Thomas M. Schindler, Anna Zak, Ramona Rorig, Kavya Chowdary Konka, Keir Hodge, Christine Crandall.

**Data curation:** Ramona Rorig, Christine Crandall.

**Formal analysis:** Thomas M. Schindler, Frank Grieger, Anna Zak, Kavya Chowdary Konka, Anna Ellsworth.

**Investigation:** Thomas M. Schindler, Anna Zak, Ramona Rorig, Kavya Chowdary Konka, Christopher Pfitzer, Keir Hodge.

**Methodology:** Thomas M. Schindler, Anna Zak, Christine Crandall.

**Project administration:** Christine Crandall.

**Resources:** Christine Crandall.

**Supervision:** Thomas M. Schindler, Christine Crandall.

**Validation:** Kavya Chowdary Konka, Anna Ellsworth.

**Visualization:** Thomas M. Schindler.

**Writing – original draft:** Thomas M. Schindler, Frank Grieger, Anna Zak, Kavya Chowdary Konka.

**Writing – review & editing:** Thomas M. Schindler, Frank Grieger, Anna Zak, Ramona Rorig, Kavya Chowdary Konka, Christopher Pfitzer, Keir Hodge, Christine Crandall.

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
