## [Decision Letter · Decision Letter 0]

21 Jan 2020

PONE-D-19-33366

Patient preferences when searching for clinical trials and adherence of study records to ClinicalTrials.gov guidance in key registry data fields

PLOS ONE

Dear Prof Schindler,

Thank you for submitting your manuscript to PLOS ONE. After careful consideration, we feel that it has merit but does not fully meet PLOS ONE’s publication criteria as it currently stands. Therefore, we invite you to submit a revised version of the manuscript that addresses the points raised during the review process.

We would appreciate receiving your revised manuscript by 20 April 2020. To enhance the reproducibility of your results, we recommend that if applicable you deposit your laboratory protocols in protocols.io, where a protocol can be assigned its own identifier (DOI) such that it can be cited independently in the future. For instructions see: http://journals.plos.org/plosone/s/submission-guidelines#loc-laboratory-protocols

We look forward to receiving your revised manuscript.

Kind regards,

Andrew Soundy

Academic Editor

PLOS ONE

2. Thank you for stating the following in the Competing Interests/Financial Disclosure* (delete as necessary) section:

"No. All authors and contributors were members of the TransCelerate Clinical Research Access and Information Exchange Initiative. The authors and contributors received support from TransCelerate in regard to project management. The funders had no role in study design, data collection and analysis, decision to publish, or preparation of the manuscript."

We note that one or more of the authors are employed by commercial companies: 'Boehringer Ingelheim Pharma', 'UCB BioSciences GmbH', 'Merck Sharp & Dohme Corp', 'Astellas Pharma', 'AbbVie Inc.', 'F. Hoffmann-La Roche Ltd' and 'GSK'.

Additional Editor Comments (if provided):

Please consider the comments made by the reviewer. Where possible use a framework to support choices e.g., https://www.equator-network.org.

Reviewers' comments:

Reviewer's Responses to Questions

**Comments to the Author**

1. Is the manuscript technically sound, and do the data support the conclusions?

Reviewer #1: Partly

2. Has the statistical analysis been performed appropriately and rigorously? 

Reviewer #1: Yes

3. Have the authors made all data underlying the findings in their manuscript fully available?

Reviewer #1: Yes

4. Is the manuscript presented in an intelligible fashion and written in standard English?

Reviewer #1: No

5. Review Comments to the Author

Reviewer #1: As the authors point out, the information contained in glinicaltrials.gov should be adapted to the needs of the patients. There are no previous studies that have evaluated this aspect, so the work is relevant, very current and may have implications for improving the information in the database.

The project, divided into several phases, has generated some interesting results. However, in my opinion, neither the structure of the work nor its contents follow the basic rules of an original article. The work has more the structure of a "report" than of a scientific article. Although it is structured in the classic sections of “introduction, material and methods”, “discussion” and “conclusions”, each of these sections mixes aspects of the rest.

Objectives: the main objectives of the study must be clearly specified at the end of the introduction, avoiding the inclusion of aspects of the methodology (first paragraph on page 5).

Methods: At the beginning of this section, the authors should give a general perspective on the methodology used, indicating the primary, secondary objectives, as well as all the analyzes that will be carried out, the rating scales included, the criteria used etc. The Validation Assessment section includes “results” that should go to the corresponding section.

Results: the reading of this section is extremely complex, excessively fragmented and it is not obvious which are the most relevant findings. In addition, each of the sections begins with a methodological description that was not previously specified in the "Methods" section. For example, the last paragraph on page 13, the last paragraph on page 14, the first paragraph on page 16, or the last paragraphs on pages 17 and 18 describe for the first time the methods used, the scales applied, etc.

Tables 4 and 5 are not easy to interpret and their contents do not seem to be aligned with the results of the text. In my opinion, the work should include fewer tables and figures, selecting the most relevant findings of the work. Although they include different analyses, some of the results in Tables 2-4 seem contradictory. For example, “location” appears as one of the most important items in table 2 and occupies a lower place in table 3. This type of inconsistencies should be explained in the Discussion.

The reason for the diversity in the rating scales and the criteria used should be explained: some range from 1 to 7, others from 1 to 8, figures 1 and 2 do not use the same credentials (one of the three most helfulp vs the item is helpful). The reasons for these variations should be explained.

Discussion: The discussion must interpret the results, however, the authors repeat again the objectives (last paragraph of page 23, second paragraph of page 24) and the methods (first paragraph of the discussion, third paragraph of page 24, second paragraph on page 25 etc.). The most important results should be discussed and interpreted, emphasizing what the study provides, the most relevant results, the most surprising, the limitations, etc.

The main results (first paragraph on page 25) should be commented at the beginning of the discussion and not on the third page. The first 3-4 paragraphs of the discussion are repetitive and do not constitute a discussion on the subject.

It seems that one of the conclusions of the study is that a short title should include information on: the condition, health measures (??), the age of the participants, the name of the drug, and the form of the drug. Is this really the relevant information for patients? Do authors think that information on the health measures used contributes more to patients engagement in research than information on start and end dates of the study? These are somewhat surprising results that the authors should comment on the discusion.

Conclusion: It is not clear which are the 2-3 main conclusions of the study.

6. PLOS authors have the option to publish the peer review history of their article (what does this mean?). If published, this will include your full peer review and any attached files.

Reviewer #1: No

---

## [Author Response · Author response to Decision Letter 0]

16 Apr 2020

In response to an email from Michelle Ellis (received 15 April 2020) we updated the funding statement and updated competing interest statements and included the revised versions in the cover letter as instructed.

---

## [Decision Letter · Decision Letter 1]

4 May 2020

Patient preferences when searching for clinical trials and adherence of study records to ClinicalTrials.gov guidance in key registry data fields

PONE-D-19-33366R1

Dear Dr. Schindler,

We are pleased to inform you that your manuscript has been judged scientifically suitable for publication and will be formally accepted for publication once it complies with all outstanding technical requirements.

With kind regards,

Andrew Soundy

Academic Editor

PLOS ONE

Additional Editor Comments (optional):

Reviewers' comments:

Reviewer's Responses to Questions

**Comments to the Author**

1. If the authors have adequately addressed your comments raised in a previous round of review and you feel that this manuscript is now acceptable for publication, you may indicate that here to bypass the “Comments to the Author” section, enter your conflict of interest statement in the “Confidential to Editor” section, and submit your "Accept" recommendation.

Reviewer #1: All comments have been addressed

2. Is the manuscript technically sound, and do the data support the conclusions?

Reviewer #1: Yes

3. Has the statistical analysis been performed appropriately and rigorously? 

Reviewer #1: Yes

4. Have the authors made all data underlying the findings in their manuscript fully available?

Reviewer #1: Yes

5. Is the manuscript presented in an intelligible fashion and written in standard English?

Reviewer #1: Yes

6. Review Comments to the Author

Reviewer #1: The authors have addressed satisfactorily to most of this reviewer's comments. The structure of the work is more in line with what is expected from an original article. The introduction section has been remarkably improved, although the last sentence “The insights gained were then to be used in the development of a tool that helps sponsors to assess the quality of their clinical trial submissions to ClinicalTrials.gov.” should be removed as it refers to the method. The results are correct, although in my opinion the number of tables is excessive. The discussion has improved very much, and it is easier to read.

7. PLOS authors have the option to publish the peer review history of their article (what does this mean?). If published, this will include your full peer review and any attached files.

Reviewer #1: No

---

## [Editor Report · Acceptance letter]

15 May 2020

PONE-D-19-33366R1 

Patient preferences when searching for clinical trials and adherence of study records to ClinicalTrials.gov guidance in key registry data fields 

Dear Dr. Schindler:

I am pleased to inform you that your manuscript has been deemed suitable for publication in PLOS ONE. Congratulations! Your manuscript is now with our production department. 

With kind regards,

on behalf of

Dr. Andrew Soundy 

Academic Editor

PLOS ONE